# Noisy Pair Corrector for Dense Retrieval

**Hang Zhang**[* 1] , **Yeyun Gong, Xingwei He**[† 2],

**Dayiheng Liu**[1] , **Daya Guo**[3] , **Jiancheng Lv**[† 1] , **Jian Guo**[† 4]

[1] College of Computer Science, Sichuan University

[1] Engineering Research Center of Machine Learning and Industry Intelligence

[2] The University of Hong Kong [3] Sun Yat-sen University [4] IDEA Research, China

hangzhang_scu@foxmail.com, hexingwei15@gmail.com

## Abstract

Most dense retrieval models contain an implicit assumption: the training query-document pairs are exactly matched. Since it is expensive to annotate the corpus manually, training pairs in real-world applications are usually collected automatically, which inevitably introduces mismatched-pair noise. In this paper, we explore an interesting and challenging problem in dense retrieval, how to train an effective model with mismatched-pair noise. To solve this problem, we propose a novel approach called **N**oisy **P**air **C**orrector (NPC), which consists of a detection module and a correction module. The detection module estimates noise pairs by calculating the perplexity between annotated positive and easy negative documents. The correction module utilizes an exponential moving average (EMA) model to provide a soft supervised signal, aiding in mitigating the effects of noise. We conduct experiments on text-retrieval benchmarks Natural Question and TriviaQA, code-search benchmarks StaQC and SO-DS. Experimental results show that NPC achieves excellent performance in handling both synthetic and realistic noise.

## 1 Introduction

With the advancements in pre-trained language models (Devlin et al., 2019; Liu et al., 2019), dense retrieval has developed rapidly in recent years. It is essential to many applications including search engine (Brickley et al., 2019), open-domain question answering (Karpukhin et al., 2020a; Zhang et al., 2021), and code intelligence (Guo et al., 2021). A typical dense retrieval model maps both queries and documents into a low-dimensional vector space and measures the relevance between them by the similarity between their respective representations (Shen et al., 2014). During training, the model utilizes query-document pairs as labelled

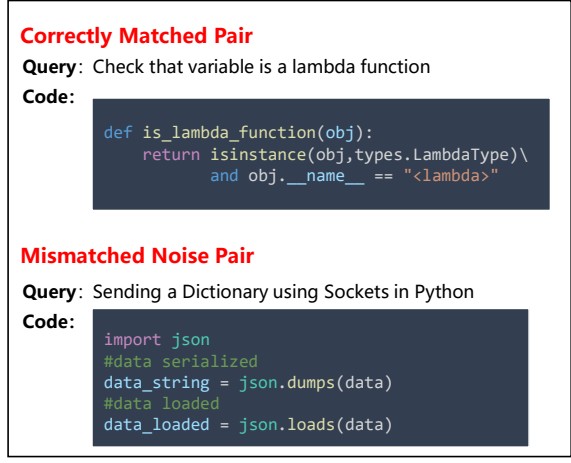

Figure 1: Two examples from StaQC training set. In the bottom example, the given code is mismatched with the query, since it can not answer the query.

training data (Xiong et al., 2021) and samples negative documents for each pair. Then the model learns to minimize the contrastive loss for obtaining a good representation ability (Zhang et al., 2022b; Qu et al., 2021).

Recent studies on dense retrieval have achieved promising results with hard negative mining (Xiong et al., 2021), pretraining (Gao and Callan, 2021a), distillation (Yang and Seo, 2020), and adversarial training (Zhang et al., 2022a). All methods contain an implicit assumption: each query is precisely aligned with the positive documents in the training set. In practical applications, this assumption becomes challenging to satisfy, particularly when the corpora is automatically collected from the internet. In such scenarios, it is inevitable that the training data will contain mismatched pairs, incorporating instances such as user mis-click noise in search engines or low-quality reply noise in Q&A communities. As shown in Fig. 1, the examples are from StaQC benchmark (Yao et al., 2018), which is automatically collected from StackOverflow. The document, i.e., code solution, can not answer the query but is incorrectly annotated as a positive doc-

---

[*]Work is done during internship at IDEA Research

[†]Corresponding author

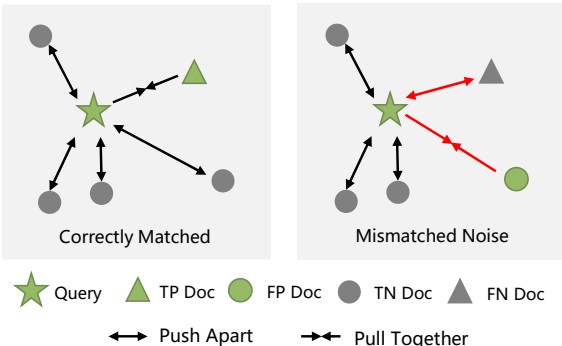

Figure 2: Effect of matched & mismatched pair for training. Green objects refer to annotated pairs, while pentagram and triangle are actually aligned pairs. In the left case, retrieval models are required to push the query with true-positive document (TP Doc) together and pull the query with true-negative documents (TN Doc) apart. In the right case, the retrieval models are misled by the mismatched data pair, where the false-positive document (FP Doc) and the false-negative document (FN Doc) are wrongly pulled and pushed, respectively.

ument. Such noisy pairs are widely present in automatically constructed datasets, which ultimately impact the performance of dense retrievers.

To train robust dense retrievers, previous works have explored addressing various types of noise. For example, RocketQA (Qu et al., 2021) and AR2 (Zhang et al., 2022a) mitigate the false-negative noise with a cross-encoder filter and distillation, respectively; coCondenser (Gao and Callan, 2021b) reduce the noise during fine-tuning with pre-training technique; RoDR (Chen et al., 2022) deal with query spelling noise with local ranking alignment. However, mismatched-pair noise (false positive problem) in dense retrieval has not been well studied. As shown in Fig. 2, mismatched-pair noise will mislead the retriever to update in the opposite direction.

Based on these observations, we propose a **Noisy Pair Corrector** (NPC) framework to solve the false-positive problem. NPC consists of noise detection and correction modules. At each epoch, the detection module estimates whether a query-document pair is mismatched by the perplexity between the annotated document and easy negative documents. Then the correction module provides a soft supervised signal for both estimated noisy data and clean data via an exponential moving average (EMA) model. Both modules are plug-and-play, which means NPC is a general training paradigm that can be easily applied to almost all retrieval models.

The contributions of this paper are as follows: (1)

We reveal and extensively explore a long-neglected problem in dense retrieval, i.e., mismatched-pair noise, which is ubiquitous in the real world. (2) We propose a simple yet effective method for training dense retrievers with mismatched-pair noise. (3) Extensive experiments on four datasets and comprehensive analyses verify the effectiveness of our method against synthetic and realistic noise. Code is available at https://github.com/hangzhang-nlp/NPC.

## 2 Preliminary

Before describing our model in detail, we first introduce the basic elements of dense retrieval, including problem definition, model architecture, and model training.

Given a query $q$, and a document collection $\mathbb{D}$, dense retrieval aims to find document $d^+$ relevant to $q$ from $\mathbb{D}$. The training set consists of a collection of query-document pairs, donated as $C = \{(q_1, d_1^+), ..., (q_N, d_N^+)\}$, where $N$ is the data size. Typical dense retrieval models adopt a dual encoder architecture to map queries and documents into a dense representation space. Then the relevance score $f(q, d)$ of query $q$ and document $d$ can be calculated with their dense representations:

$$f_\theta(q, d) = sim\left(E(q; \theta), E(d; \theta)\right), \quad (1)$$

where $E(\cdot; \theta)$ denotes the encoder module parameterized with $\theta$, and $sim$ is the similarity function, e.g., euclidean distance, cosine distance, inner-product. Existing methods generally leverage the approximate nearest neighbor technique (ANN) (Johnson et al., 2019) for efficient search.

For training dense retrievers, conventional approaches leverage contrastive learning techniques (Karpukhin et al., 2020a; Zhang et al., 2022b). Given a training pair $(q_i, d_i^+) \in C$, these methods sample $m$ negative documents $\{d_{i,1}^-, ..., d_{i,m}^-\}$ from a large document collection $\mathbb{D}$. The retriever's objective is to minimize the contrastive loss, pushing the similarity of positive pairs higher than negative pairs. Previous work (Xiong et al., 2021) has verified the effectiveness of the negative sampling strategy. Two commonly employed strategies are "In-Batch Negative" and "Hard Negative" (Karpukhin et al., 2020a; Qu et al., 2021).

The above training paradigm assumes that the query-document pair $(q_i, d_i^+)$ in training set $C$ is correctly aligned. However, this assumption is difficult to satisfy in real-world applications (Qu et al.,

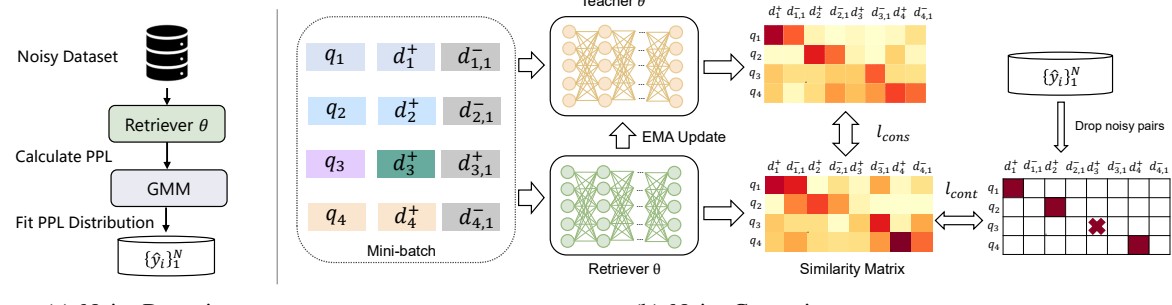

| (a) Noise Detection | (b) Noise Correction |

Figure 3: Overview of noise detection and noise correction. (a) Procedure of Noise Detection. At each epoch, we first calculate the perplexity of all training query-document pairs using the retriever $\theta$; next fit the perplexity distribution with Gaussian Mixture Model to get the correctly matched probability of each pair; finally estimate the flag set $\{\hat{y}_i\}_{i=1}^{N}$ by setting the threshold. (b) Framework of Noise Correction. Given a batch of data pairs, where $d_{i,1}^{-}$ is the hard negative of $q_i$ and $\{q_3, d_3^{+}\}$ is the estimated noisy pair, the retriever $\theta$ and teacher $\theta^*$ compute similarity matrices $S_\theta$ and $S_{\theta*}$ for all queries and documents, respectively. The retriever learns to minimize (1) $L_{cont}$: the negative likelihood probability of true positive documents; (2) $L_{cons}$: the KL divergence between $S_\theta$ and the rectified soft label $S_{\theta*}$ after normalization.

2021; Li et al., 2022; Wang et al., 2022). In practice, most training data pairs are collected automatically without manual inspection, such as inevitably leading to the inclusion of some mismatched pairs.

## 3 Method

We propose NPC framework to learn retrievers with mismatched-pair noise. As shown in Fig. 3, NPC consists of two parts: (a) the noise detection module as described in Sec. 3.1, and (b) the noise correction module as described in Sec. 3.2.

### 3.1 Noise Detection

The noise detection module is meant to detect mismatched pairs in the training set. We hypothesize that: dense retrievers will first learn to distinguish correctly matched pairs from easy negatives, and then gradually overfit the mismatched pairs. Therefore, we determine whether a training pair is mismatched by the perplexity between the annotated document and easy negative documents.

Specifically, given a retriever $\theta$ equipped with preliminary retrieval capabilities, and an uncertain pair $(q_i, d_i)$, we calculate the perplexity as follows:

$$PPL_{(q_i,d_i,\theta)} = -\log \frac{e^{\tau f_\theta(q_i,d_i)}}{e^{\tau f_\theta(q_i,d_i)} + \sum_{j=1}^{m} e^{\tau f_\theta(q_i,d_{i,j}^{-})}}, \quad (2)$$

where $\tau$ is a hyper-parameter, $d_{i,j}^{-}$ is the negative document randomly sampled from the document collection $\mathbb{D}$. Note that $d_{i,j}^{-}$ is a randomly selected negative document, not a hard negative. We discuss

this further in Appendix C. In practice, we adopt the "In-Batch Negative" strategy for efficiency.

After obtaining the perplexity of each pair, an automated method is necessary to differentiate between the noise and the clean data. We note that there is a bimodal effect between the distribution of clean samples and the distribution of noisy samples. An example can be seen in Figure 4(b). Motivated by this, we fit the perplexity distribution over all training pairs with a two-component Gaussian Mixture Model (GMM):

$$p\left(PPL \mid \theta\right) = \sum_{k=1}^{K} \pi_k \phi\left(PPL \mid k\right), \quad (3)$$

where $\pi_k$ and $\phi\left(PPL \mid k\right)$ are the mixture coefficient and the probability density of the $k$-th component, respectively. We optimize the GMM with the Expectation-Maximization algorithm (Dempster et al., 1977).

Based on the above hypothesis, we treat training pairs with higher $PPL$ as noise and those with lower $PPL$ as clean data. So the estimated clean flag can be calculated as follows:

$$\hat{y}_i = \mathbb{I}\left(p(\kappa \mid PPL_{(q_i,d_i,\theta)}) > \lambda\right), \quad (4)$$

where $\hat{y}_i \in \{1, 0\}$ denotes whether we estimate the pair $(q_i, d_i)$ to be correctly matched or not, $\kappa$ is the GMM component with the lower mean, $\lambda$ is the threshold. $p(\kappa \mid PPL_{(q_i,d_i,\theta)})$ is the posterior probability over the component $\kappa$, which can be intuitively understood as the correctly annotated con-

fidence. We set $\lambda$ to 0.5 in all experiments. Note that before noise detection, the retriever should equip with preliminary retrieval capabilities. This can be achieved by initializing it with a strong unsupervised retriever or by pre-training it on the entire noise dataset.

## 3.2 Noise Correction

Next, we will introduce how to reduce the impact of noise pairs after obtaining the estimated flag set $\{\hat{y}_i\}_{i=1}^N$. One quick fix is to discard the noise data directly, which is sub-optimal since it wastes the query data in noisy pairs. In this work, we adopt a self-ensemble teacher to provide rectified soft labels for noisy pairs. The teacher is an exponential moving average (EMA) of the retriever, and the retriever is trained with a weight-averaged consistency target on noisy data.

Specifically, given a retriever $\theta$, the teacher $\theta^*$ is updated with an exponential moving average strategy as follows:

$$\theta_t^* = \alpha\theta_{t-1}^\star + (1-\alpha)\theta_t, \qquad (5)$$

where $\alpha$ is a momentum coefficient. Only the parameters $\theta$ are updated by back-propagation.

For a query $q_i$ and the candidate document set $D_{q_i}$, where $D_{q_i} = \{d_{i,j}\}_{j=1}^m$ could consist of annotated documents, hard negatives and in-batch negatives, we first get teacher's and retriever's similarity scores, respectively. Then, the retriever $\theta$ is expected to keep consistent with its smooth teacher $\theta^*$. To achieve this goal, we update the retriever $\theta$ by minimizing the KL divergence between the student's distribution and the teacher's distribution.

To be concrete, the similarity scores between $q_i$ and $D_{q_i}$ are normalized into the following distributions:

$$p_\phi(d_{i,j}|q_i; D_{q_i}) = \frac{e^{\tau f_\phi(q_i, d_{i,j})}}{\sum_{j=1}^m e^{\tau f_\phi(q_i, d_{i,j})}}, \phi \in \{\theta, \theta^*\}, \qquad (6)$$

Then, the consistency loss $L_{cons}$ can be written as:

$$L_{cons} = KL(p_\theta(.|q_i; D_{q_i}), p_{\theta^*}(.|q_i; D_{q_i})), \quad (7)$$

where $KL(\cdot)$ is the KL divergence, $p_\theta(.|q_i; D_{q_i})$ and $p_{\theta^*}(.|q_i; D_{q_i})$ denote the conditional probabilities of candidate documents $D_{q_i}$ by the retriever $\theta$ and the teacher $\theta^*$, respectively.

For the estimated noisy pair, the teacher corrects the supervised signal into a soft label. For the

---

**Algorithm 1** Noisy Pair Corrector (NPC)

**Require:** Retriever $\theta$; Noisy Training dataset $C$.
1: Warm up the retriever $\theta$.
2: Initial EMA model $\theta^*$ with $\theta$;
3: **for** $i = 1 : num\_epoch$ **do**
4:     Calculate PPL of training pairs with random negatives using Eq.2;
5:     Fit PPL distribution with GMM;
6:     Get the estimated flag set $\{\hat{y}_i\}$ using Eq.4;
7:     **for** $i = 1 : num\_batch$ **do**
8:         Sample negatives with "In-Batch Negative" or "Hard Negative" strategy;
9:         Calculate rectified soft labels with EMA model $\theta^*$;
10:        Train $\theta$ by optimizing Eq.8;
11:        Update EMA model $\theta^*$ using Eq.5;
12:     **end for**
13: **end for**

---

estimated clean pair, we calculate the contrastive loss and consistency loss. So the overall loss is formalized:

$$L = \hat{y}_i L_{cont} + L_{cons}, \qquad (8)$$

where $\hat{y}_i \in \{1, 0\}$ is estimated by the noise detection module.

## 3.3 Overall Procedure

NPC is a general training framework that can be easily applied to most retrieval methods. Under the classical training process of dense retrieval, We first warmup the retriever with the typical contrastive learning method to provide it with basic retrieval abilities, and then add the noise detection module before training each epoch and the noise correction module during training. The detail is presented in Algorithm 1.

# 4 Experiments

## 4.1 Datasets

To verify the effectiveness of NPC in robust dense retrieval, we conduct experiments on four commonly-used benchmarks, including Natural Questions (Kwiatkowski et al., 2019), Trivia QA (Joshi et al., 2017), StaQC (Yao et al., 2018) and SO-DS (Heyman and Van Cutsem, 2020).

StaQC is a large dataset that collects real query-code pairs from Stack Overflow*. The dataset has

---
*https://stackoverflow.com/

| Methods | StaQC | | | SO-DS | | |
|---|---|---|---|---|---|---|
| | R@3 | R@10 | MRR | R@3 | R@10 | MRR |
| BM25$_{desc}$ (Heyman and Van Cutsem, 2020) | 8.0 | 13.3 | 7.5 | 23.8 | 32.3 | 21.6 |
| NBOW (Heyman and Van Cutsem, 2020) | 10.9 | 16.6 | 9.5 | 27.7 | 38.0 | 24.7 |
| USE (Heyman and Van Cutsem, 2020) | 12.8 | 20.3 | 11.7 | 33.3 | 48.5 | 30.4 |
| CodeBERT (Feng et al., 2020) | - | - | 23.4 | - | - | 23.1 |
| GraphCodeBERT (Guo et al., 2021) | - | - | 24.1 | - | - | 25.2 |
| CodeRetriever (In-Batch Negative) (Li et al., 2022) | - | - | 25.5 | - | - | 27.1 |
| CodeRetriever (Hard Negative) (Li et al., 2022) | - | - | 24.6 | - | - | 31.8 |
| UniXcoder (In-Batch Negative) (Guo et al., 2022) | 29.98 | 47.47 | 28.04 | 31.90 | 51.21 | 28.29 |
| UniXcoder (Hard Negative) (Guo et al., 2022) | 31.18 | 48.38 | 28.63 | 33.42 | 53.37 | 29.97 |
| NPC (In-Batch Negative) | 33.07 | 50.35 | 30.39 | 35.58 | 54.54 | 30.96 |
| NPC (Hard Negative) | **34.38** | **52.20** | **31.36** | **38.00** | **56.51** | **32.49** |

Table 1: Retrieval performance on StaQC and SO-DS, which are realistic-noisy datasets. The results of the first block are mainly borrowed from published papers (Heyman and Van Cutsem, 2020; Li et al., 2022). If the results are not provided, we mark them as "-".

been widely used on code summarization (Peddamail et al., 2018) and code search (Heyman and Van Cutsem, 2020). SO-DS mines query-code pairs from the most upvoted Stack Overflow posts, mainly focuses on the data science domain. Following previous works (Heyman and Van Cutsem, 2020; Li et al., 2022), we resort to Recall of top-k (R@k) and Mean Reciprocal Rank (MRR) as the evaluation metric. StaQC and SO-DS are constructed automatically without human annotation. Therefore, there are numerous mismatched pairs in training data.

Natural Questions (NQ) collects real queries from the Google search engine. Each question is paired with an answer span and golden passages from the Wikipedia pages. Trivia QA (TQ) is a reading comprehension dataset authored by trivia enthusiasts. During the retrieval stage of both datasets, the objective is to identify positive passages from a large collection. Positive pairs in these datasets are assessed based on strict rule, i.e., whether passages contain answers or not (Karpukhin et al., 2020a). Consequently, we consider these datasets to be of high quality. Thus, we leverage them for simulation experiments to quantitatively analyze the impact of varying proportions of noise. Drawing inspiration from the setting in the noisy classification task (Han et al., 2018), we simulate the mismatched-pair noise by randomly pairing queries with unrelated documents.

## 4.2 Implementation Details

NPC is a general training paradigm that can be directly applied to almost all retrieval models. For

StaQC and SO-DS, we adopt UniXcoder (Guo et al., 2022) as our backbone, which is the SoTA model for code representation. Following Guo et al. (2022), we adopt the cosine distance as a similarity function and set temperature $\lambda$ to 20. We update model parameters using the Adam optimizer and perform early stopping on the development set. The learning rate, batch size, warmup epoch, and training epoch are set to 2e-5, 256, 5, and 10, respectively. In the "Hard Negative" setting, we adopt the same strategy as Li et al. (2022).

For NQ and TQ, we adopt BERT (Devlin et al., 2019) as our initial model. Following Karpukhin et al. (2020a), we adopt inner-product as the similarity function and set temperature $\lambda$ to 1. The max sequence length is 16 for query and 128 for passage. The learning rate, batch size, warmup epoch, and training epoch are set to 2e-5, 512, 10, and 40, respectively. We adopt "BM25 Negative" and "Hard Negative" strategies as described in the DPR toolkit [†]. For a fair comparison, we implement DPR (Karpukhin et al., 2020a) with the same hyper-parameters. All experiments are run on 8 NVIDIA Tesla A100 GPUs. The implementation of NPC is based on Huggingface (Wolf et al., 2020).

## 4.3 Results

**Results on StaQC and SO-DS:** Table 1 shows the results on the realistic-noisy datasets StaQC and SO-DS. Both datasets contain a large number of real noise pairs. The first block shows the results of previous SoTA methods. BM25$_{desc}$ is a traditional

---

[†] https://github.com/facebookresearch/DPR

| Noisy | Methods | Natural Questions | | | | Trivia QA | | | |
|---|---|---|---|---|---|---|---|---|---|
| | | R@1 | R@5 | R@20 | R@100 | R@1 | R@5 | R@20 | R@100 |
| | BM25∗ | - | - | 59.1 | 73.7 | - | - | 66.9 | 76.7 |
| | DPR∗ | - | - | 78.4 | 85.4 | - | - | 79.4 | 85.0 |
| 20 | DPR (BM25 Negative) | 27.07 | 47.79 | 63.36 | 75.69 | 35.73 | 52.88 | 64.05 | 74.16 |
| | coCondenser (BM25 Negative) | 29.12 | 51.02 | 67.45 | 77.93 | 39.41 | 56.72 | 67.34 | 76.04 |
| | Co-teaching (BM25 Negative) | 26.02 | 52.48 | 63.46 | 76.11 | 28.65 | 53.01 | 64.99 | 74.05 |
| | DPR-C (BM25 Negative) | 43.69 | 66.62 | 79.07 | 86.12 | 52.10 | 70.52 | 79.05 | 85.08 |
| | NPC (BM25 Negative) | 45.22 | 68.42 | 79.76 | 86.56 | 52.34 | 70.22 | 79.10 | 84.86 |
| | DPR (Hard Negative) | 37.61 | 60.73 | 71.68 | 79.56 | 43.39 | 60.67 | 70.34 | 77.88 |
| | coCondenser (Hard Negative) | 40.71 | 63.41 | 74.33 | 81.22 | 47.42 | 64.80 | 73.38 | 80.07 |
| | RocketQA (Hard Negative) | 43.32 | 64.25 | 74.96 | 81.42 | 49.90 | 65.72 | 74.04 | 80.39 |
| | Co-teaching (Hard Negative) | 31.78 | 56.32 | 66.12 | 77.56 | 33.28 | 57.29 | 66.50 | 75.62 |
| | DPR-C (Hard Negative) | 51.66 | 72.40 | 81.50 | 87.80 | 55.35 | 72.36 | 80.33 | 85.34 |
| | NPC (Hard Negative) | **51.85** | **73.06** | **82.47** | **87.80** | **56.03** | **72.54** | **80.59** | **85.58** |
| 50 | DPR (BM25 Negative) | 16.12 | 33.88 | 49.70 | 63.38 | 20.09 | 34.63 | 47.42 | 61.04 |
| | coCondenser (BM25 Negative) | 18.28 | 36.37 | 52.01 | 65.92 | 22.80 | 38.01 | 51.00 | 63.79 |
| | Co-teaching (BM25 Negative) | 23.72 | 50.32 | 64.86 | 74.92 | 26.56 | 51.22 | 63.78 | 73.77 |
| | DPR-C (BM25 Negative) | 41.29 | 65.21 | 78.48 | 85.70 | 49.61 | 68.81 | 78.00 | 84.23 |
| | NPC (BM25 Negative) | 42.87 | 65.65 | 78.37 | 85.76 | 50.80 | 68.98 | 78.21 | 84.43 |
| | DPR (Hard Negative) | 23.87 | 42.34 | 55.12 | 67.06 | 28.47 | 45.12 | 56.88 | 67.62 |
| | coCondenser (Hard Negative) | 24.55 | 44.16 | 56.69 | 68.72 | 31.05 | 47.81 | 59.48 | 70.14 |
| | RocketQA (Hard Negative) | 26.83 | 45.72 | 57.32 | 69.24 | 33.67 | 49.28 | 60.32 | 70.46 |
| | Co-teaching (Hard Negative) | 30.12 | 55.94 | 65.81 | 76.90 | 31.85 | 55.37 | 65.29 | 75.02 |
| | DPR-C (Hard Negative) | **48.87** | 70.52 | **81.44** | 87.17 | 53.07 | **70.36** | 79.02 | 84.69 |
| | NPC (Hard Negative) | 48.81 | **70.60** | 81.17 | **87.20** | **53.09** | 70.27 | **79.31** | **84.96** |

Table 2: Retrieval performance on Natural Questions and Trivia QA under the noise ratio of 20%, and 50%, respectively. The results of BM25∗ and DPR∗ are borrowed from Karpukhin et al. (2020a). If the results are not provided, we mark them as "-".

sparse retriever based on the exact term matching of queries and code descriptions. NBOW is an unsupervised retriever that leverages pre-trained word embedding of queries and code descriptions for retrieval. USE is a simple dense retriever based on transformer. CodeBERT, GraphCodeBERT are pre-trained models for code understanding using large-scale code corpus. CodeRetriever is a pre-trained model dedicated to code retrieval, which is pre-trained with unimodal and bimodal contrastive learning on a large-scale corpus. UniXcoder is also a pretrained model that utilizes multi-modal data, including code, comment, and AST, for better code representation. The results are implemented by ourselves for a fair comparison with NPC. The bottom block shows the results of NPC using two negative sampling strategies.

From the results, we can see that our proposed NPC consistently performs better than the evaluated models across all metrics. Compared with the strong baseline UniXcoder which ignores the mismatched-pair problem, NPC achieves a significant improvement with both "in-batch negative" and "hard negative" sampling strategies. It indi-

cates that the mismatched-pair noise greatly limits the performance of dense retrieval models, and NPC can mitigate this negative effect. We also show some noisy examples detected by NPC in Appendix A.

**Results on NQ and TQ:** Table 2 shows the results on the synthetic-noisy datasets NQ and TQ under the noise ratio of 20%, and 50%. We compare NPC with BM25 (Yang et al., 2017) and DPR (Karpukhin et al., 2020a). BM25 is an unsupervised sparse retriever that is not affected by noisy data. DPR (Karpukhin et al., 2020a) is a widely used method for training dense retrievers. coCondenser (Gao and Callan, 2021b) leverage pre-training to enhance models' robustness. RocketQA (Qu et al., 2021) adopts a cross-encoder to filter false negatives in the "Hard Negative" strategy. Co-teaching (Han et al., 2018) uses the samples with small loss to iteratively train two networks, which is widely used in the noisy label classification task. We implement baselines using two negative sampling strategies. Besides, we evaluate DPR on clean datasets by discarding the synthetic-noisy pairs, denoted by DPR-C. DPR-C is a strong

| Methods | | | NQ | | StaQC | |
|---|---|---|---|---|---|---|
| De | Co | HN | R@20 | R@100 | R@1 | R@3 |
| - | - | - | 48.22 | 62.31 | 18.08 | 31.09 |
| - | ✓ | - | 55.90 | 69.33 | 18.51 | 31.01 |
| ✓ | - | - | 75.19 | 83.31 | 20.05 | 32.71 |
| ✓ | ✓ | - | 77.50 | 84.79 | 20.70 | 33.55 |
| - | - | ✓ | 54.63 | 65.54 | 18.66 | 31.74 |
| - | ✓ | ✓ | 58.63 | 69.06 | 19.35 | 32.09 |
| ✓ | - | ✓ | 77.59 | 85.03 | 20.93 | 33.55 |
| ✓ | ✓ | ✓ | **80.07** | **85.89** | **21.93** | **34.51** |

Table 3: Ablation studies on StaQC dev set and NQ dev set under noise ratio of 50%.

baseline that is not affected by mismatched pairs.

We can observe that (1) As the noise ratio increases, DPR, coCondenser, and RocketQA experience a significant decrease in performance. At a noise rate of 50%, they perform worse than unsupervised BM25. (2) Despite Co-teaching having good noise resistance, its performance is still low. This indicates that methods for dealing with label noise in classification are not effective for retrieval. (3) NPC outperforms baselines by a large margin, with only a slight performance drop when the noise increases. Even comparing DPR-C, NPC still achieves competitive results.

## 4.4 Analysis

**Ablations of Noise Detection and Noise Correction:** To get a better insight into NPC, we conduct ablation studies on the realistic-noisy dataset StaQC and the synthetic-noisy dataset NQ under the noise ratio of 50%. The results are shown in Table 3. "De" and "Co" refer to noise detection and noise correction, respectively. "HN" indicates whether to perform "Hard Negative" strategy. For both synthetic noise and realistic noise, we can see that the noise detection module brings a significant gain, no matter which negative sampling strategy is used. Correction also enhances the robustness of the retriever since it provides rectified soft labels which can lead the model output to be smoother. The results show that combining the two obtains better performance compared with only using the detection module or correction module.

**Impact of Warmup Epoch:** According to the foregoing, NPC starts by warming up. In Table 4, we pre-training the retriever on the noisy dataset for warming up, and show the performance of NPC with different various epoch numbers $n$. In this

| Setting | R@1 | R@5 | R@20 | R@100 |
|---|---|---|---|---|
| $n=1$ | 50.58 | 69.93 | 79.87 | 84.96 |
| $n=5$ | 50.03 | 69.64 | 80.17 | 85.76 |
| $n=10$ | 50.07 | 69.93 | 80.07 | 85.89 |
| $n=20$ | 38.09 | 60.31 | 72.00 | 80.07 |

Table 4: Performance of NPC on NQ dev set with different warmup epoch number $n$.

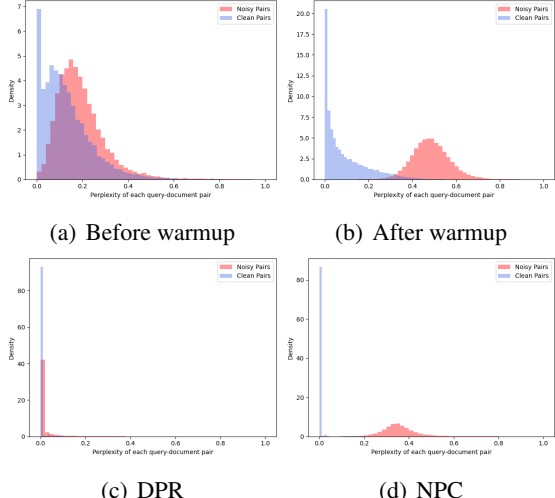

(a) Before warmup   (b) After warmup

(c) DPR   (d) NPC

Figure 4: Perplexity distribution of training pairs under different settings.

experiment, we adopt "Hard Negative" sampling strategy. We find that NPC achieves good results when the warmup epoch is relatively small $(1-10)$. However, when the warmup epoch is too large, the performance will degrade. We believe that a prolonged warmup causes overfitting to noise samples.

**Impact of Iterative Detection:** In the training of NPC, we perform iterative noise detection every epoch. A straightforward approach is to detect the noise only once after warmup and fix the estimated flag set $\{\hat{y}_i\}$. To study the effectiveness of iterative detection, we conducted an ablation study. The results are shown in Table 5. We can see that the model performance degrades after removing iterative detection.

**Ablations of PPL:** We distinguish noise pairs according to the perplexity between the annotated positive document and easy negatives. When calculating the perplexity, "Hard Negative" will cause trouble for detection. We construct ablation experiments to verify this, and the results are shown in Table 5. We can see that the perplexity with "Hard Negative" results in performance degradation.

**Visualization of Perplexity Distribution:** In Fig. 4, we illustrate the perplexity distribution of

| Setting | R@1 | R@5 | R@20 | R@100 |
|---|---|---|---|---|
| NPC | 50.07 | 69.93 | 80.07 | 85.89 |
| *-w/o iterative detection* | 47.29 | 68.39 | 78.79 | 85.38 |
| *-ppl with HN* | 42.81 | 65.06 | 75.22 | 83.09 |

Table 5: Ablation studies of iterative noise detection and perplexity variants

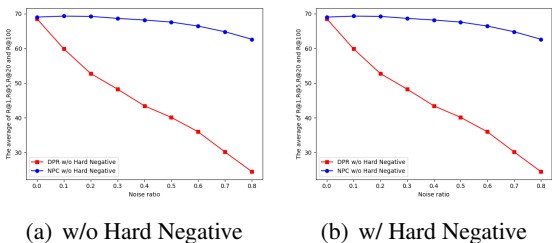

(a) w/o Hard Negative          (b) w/ Hard Negative

Figure 5: Retrieval performance of DPR and NPC on NQ dev set under different noise ratios.

training pairs before and after warmup, after training with DPR, and after training with NPC. The experiment is on NQ under the noise ratio of 50%. We can see that the perplexity of most noisy pairs is larger than the clean pairs after warmup, which verifies our hypothesis in Sec. 3.1. Comparing Fig. 4(c) and Fig. 4(d), we find that the retriever trained with DPR will overfit the noise pairs. However, NPC enables the retriever to correctly distinguish clean and noisy pairs because it avoids the dominant effect of noise during network optimization.

**Analysis of Generalizability** Fig. 5 shows the performance of DPR and NPC under the noise ratio ranging from 0% to 80%. We can see that as the noise ratio increases, the performance degradation of DPR is much larger than that of NPC, which demonstrates the generalizability of NPC. Furthermore, even though NPC is designed to deal with mismatched-pair noise, it achieves competitive results when used in a noise-free setting.

# 5 Related Work

## 5.1 Dense Retrieval

Dense retrieval has shown better performance than traditional sparse retrieval methods (Lee et al., 2019; Karpukhin et al., 2020a). The studies of dense retrieval can be divided into two categories, (1) unsupervised pre-training to get better initialization and (2) more effective fine-tuning on labeled data. In the first category, some researchers focus on how to generate contrastive pairs automatically from a large unsupervised corpus (Lee et al., 2019; Chang et al., 2019; Ma et al., 2022; Li et al., 2022). Another line of research enforces

the model to produce an information-rich CLS representation (Gao and Callan, 2021a,b; Lu et al., 2021). As for effective fine-tuning strategies (He et al., 2022b), recent studies show that negative sampling techniques are critical to the performance of dense retrievers. DPR (Karpukhin et al., 2020b) adopts in-batch negatives and BM25 negatives; ANCE (Xiong et al., 2021), RocketQA (Qu et al., 2021), and AR2 (Zhang et al., 2022a) improve the hard negative sampling by iterative replacement, denoising, and adversarial framework, respectively. Several works distill knowledge from ranker to retriever (Izacard and Grave, 2020; Yang and Seo, 2020; Ren et al., 2021; Zeng et al., 2022). Some studies incorporate lexical-aware sparse retrievers to convey lexical-related knowledge to dense retrievers, thereby enhancing the dense retriever's ability to recognize lexical matches (Shen et al., 2023; Zhang et al., 2023).

Although the above methods have achieved promising results, they are highly dependent on correctly matched data, which is difficult to satisfy in real scenes. The mismatched-pair noise problem has seldom been considered. Besides, some studies utilize large-sized generative models (He et al., 2023) to guide retrievers, which achieve impressive performance without paired data (Sachan et al., 2022, 2021; Gao et al., 2022; He et al., 2022a). Although these models exhibit some robustness to noisy data, their success depends on the availability of strong generative models. Moreover, their applicability will be limited in domains where generative models do not perform well.

## 5.2 Denoising Techniques

One related task to our work is *Noisy Label*. Numerous methods have been proposed to solve this problem, and most of them focus on the classification task (Han et al., 2020). Some works design robust loss functions to mitigate label noise (Ghosh et al., 2017; Ma et al., 2020). Another line of work aims to identify noise from the training set with the memorization effect of neural networks (Silva et al., 2022; Liang et al., 2022; Bai et al., 2021).

These studies mainly focus on classification. NPC studies the mismatched noise problem in dense retrieval rather than the noise in category annotations, which is more complex to handle. Several pre-training approaches noticed the problem of mismatched noisy pairs. ALIGN (Jia et al., 2021) and CLIP (Radford et al., 2021) claim that utilizing

large-scale image-text pairs can ignore the existence of noise. E5 (Wang et al., 2022) employs a consistency-based rule to filter the pre-training data. Although they slightly realized the existence of noisy pairs during pre-train, none of them give a specialized solution to solve it and extensively explored the characteristics of noisy text pairs. Some recent works (Huang et al., 2021; Han et al., 2023) study the noisy correspondence problem in cross-modal retrieval. Although the "mismatched-pair noisy" problem in cross-modal retrieval and text retrieval shares similarities, the specific settings and methods used in these two areas are notably distinct. it is challenging to directly apply these cross-modal retrieval works to document and code retrieval. Our NPC is the first systematic work to explore mismatched-pair noise in document/code retrieval.

## 6   Conclusion

This paper explores a neglected problem in dense retrieval, i.e., mismatched-pair noise. To solve this problem, we propose a generalized Noisy Pair Corrector(NPC) framework, which iteratively detects noisy pairs per epoch based on the perplexity and then provides rectified soft labels via an EMA model. The experimental results and analysis demonstrate the effectiveness of NPC in effectively handling both synthetic and realistic mismatched-pair noise.

## Limitations

This work mainly focuses on training the dense retrieval models with mismatched noise. There may be two possible limitations in our study.

1) Due to the limited computing infrastructure, we only verified the robustness performance of NPC based on the classical retriever training framework. We leave experiments to combine NPC with more effective retriever training methods such as distillation (Ren et al., 2021), AR2 (Zhang et al., 2022a), as future work.

2) Mismatched-pair noise may also exist in other tasks, such as recommender systems. We will consider extending NPC to more tasks.

## Acknowledgement

This work is supported by the Fundamental Research Funds for the Central Universities under Grant 1082204112364 and the Key Program of the National Science Foundation of China under Grant 61836006.

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

# A  Qualitative Analysis

Table 9 lists some mismatched pairs detected by NPC in StaQC training set. We can see that these mismatched pairs are almost irrelevant and can be correctly detected by NPC. These examples are not well aligned, mainly due to the low-quality answers of the open community (cases 2 and 4), inappropriate data preprocessing in the collection phase (cases 2 and 3), and other reasons. It is well known that collecting and cleaning training data is expensive and complex work. Automatically constructed datasets in real-world applications often contain such mismatched-pair noise. Our method can mitigate the impact caused by such noise during training.

# B  Statistics of Datasets

The statistics of datasets are shown in Table 6.

| Dataset | Train | Dev | Test | Corpus size |
|---------|-------|------|-------|-------------|
| StaQC | 203.7K | 2.6K | 2.7K | 14.6K |
| SO-DS | 12.1K | 0.9K | 1.1K | 12.1K |
| NQ | 79.2K | 8,8K | 3.6K | 21 M |
| TQ | 78.8K | 8.8k | 11.3K | 21 M |

Table 6: The statistics of datasets. Corpus size means the size of document corpus for evaluation.

# C  Discussion about Perplexity

We calculate the perplexity between the annotated document and easy negative documents during noise detection. We emphasize that the negative documents are randomly selected from the document collection $\mathbb{D}$. It is not suitable to adopt "Hard Negative" sampling strategy when calculating the perplexity. Although hard negatives are important to train a strong dense retriever, they will cause trouble during noise detection. Specifically, it is expected that the retriever is confused only between false positive and negative documents and can confidently distinguish true positive and negative documents. But if we adopt "Hard Negative" when calculating the perplexity, the retriever will also be confused between true positive and hard negative documents, which will affect noise detection. We construct ablation experiments to verify this, and the results are shown in Table 5.

# D  Integration with stronger methods

We conducted comprehensive experiments that integrated NPC into coCondenser and RocketQAv2. The subsequent experiments were conducted on the NQ dataset with a 50% noise ratio. We first combined NPC with coCondenser which is a pretrained model specialized for dense retrieval tasks. The results are shown in Table 7

|  | R@1 | R@5 | R@20 | R@100 |
|--|-----|-----|------|-------|
| BM25 Negative | | | | |
| coCondenser | 18.28 | 36.37 | 52.01 | 65.92 |
| coCondenser-C | 44.75 | 68.91 | 80.89 | 87.31 |
| coCondenser+NPC | 47.31 | 70.38 | 81.58 | 87.38 |
| Hard Negative | | | | |
| coCondenser | 24.55 | 44.16 | 56.69 | 68.72 |
| coCondenser-C | 49.31 | 71.99 | 82.41 | 88.38 |
| coCondenser+NPC | 50.66 | 72.42 | 82.64 | 88.31 |

Table 7: Retrieval performance on NQ after combining NPC with coCondenser.

It's evident that NPC significantly enhances the robustness of coCondenser against noise associated with mismatched pairs. This observation underscores the compatibility between NPC and pretrained dense retrievers.

Furthermore, we combined NPC with RocketQAv2 which adopted a cross-encoder as a teacher and dynamically distilled knowledge to the dense retriever. The results are shown in Table 8:

|  | R@1 | R@5 | R@20 | R@100 |
|--|-----|-----|------|-------|
| RocketQAv2 | 32.30 | 51.37 | 62.19 | 70.79 |
| RocketQAv2-C | 52.63 | 73.51 | 83.21 | 88.71 |
| RocketQAv2+NPC | 52.59 | 73.83 | 83.32 | 88.69 |

Table 8: Retrieval performance on NQ after combining NPC with RocketQAv2.

To combine NPC with RocketQAv2, we integrate the noise detection and the correction modules in each training epoch of RocketQAv2. From the table, we can find that although RocketQAv2 uses a powerful cross-encoder as a teacher, it is still limited by the noise of the training data and shows low performance. NPC can effectively harmonize with RocketQAv2 to mitigate the problems caused by mismatched pair noise. We will add these experiment results to the next version.

| | Question | Code |
|---|---|---|
| 1 | Split words in a nested list into letters | » [list(l[0]) for l in mylist] |
| 2 | Dictionary in python problem | » s = problem.getSuccessors( getStartState()) |
| 3 | Find the Common first name from Django Auth user Model | » import operator |
| 4 | Find all text files not containing some text string | » lst = [1,2,4,6,3,8,0,5]
» for n in lst[:]:
»» if n % 2 == 0:
»»» lst.remove(n)
» lst |

Table 9: Some noisy pairs detected by NPC in StaQC training set.