# OpenReview forum: "Noisy Pair Corrector for Dense Retrieval"
_EMNLP/2023/Conference — EMNLP 2023 Findings_

### Official Review · Reviewer_YGVa · 2023-08-03

**Soundness:** 4

**Excitement:**

4: Strong: This paper deepens the understanding of some phenomenon or lowers the barriers to an existing research direction.

**Missing References:**

See the "Reasons To Reject" part

**Paper Topic And Main Contributions:**

This paper proposes a method called Noisy Pair Corrector for eliminating the effect of pseudo-positive examples on the fine-tuning step of dense retrieval fine-tuning. In principle, this method is simple enough to be theoretically applied to current dense retrieval models of multiple architectures. The experimental results show it has a large improvement in the data in a variety of areas.

**Questions For The Authors:**

Q1: What is a simple negative sample? The text does not state its definition. Is a negative sample a simple negative sample if it is not a hard negative sample?

Q2: What is the exact expression of $L_{cont}$? I only found its textual description in Figure 3

Q3: Are the results in the paper new SOTA? This part is too vague in its description

**Reasons To Accept:**

1. Too little research has been done on the problem of mismatched-pair noise, and this article is a keen probe into this pervasive and unresolved problem in real-world data.

2. The experimental part of the article is well-detailed and shows the reader the effectiveness of the method in various aspects.

3. The experimental results have been shown to be valid in a variety of task settings, demonstrating the applicability of this method in different domains

**Reasons To Reject:**

1. I believe the first point of the contribution overstates the contribution of this paper. As far as I know, this paper may be the first study of the "mismatched-pair noise" problem in the field of document retrieval and code retrieval, but similar work already exists on other dense retrieval tasks, e.g., [1.2].
2. The method proposed in this paper is called "Noisy Pair Corrector", but I think NPC does not achieve the purpose of correction. According to the results in Table 2, even though the proportion of noisy labeled pairs has come to 50%, the gap between the NPC results and the DPR-C results is weak, which seems to imply that the supervisory signals of Eq. 7 have almost no improvement on the results.
3. The different sections of this paper are not organized closely enough. On the one hand, the "warmup" and "iterative detection" of the experiment section, which are described at great length, are not even mentioned before the experiment section, and I can't even fully understand what they mean. On the other hand, the paper highlights the plug-and-play nature of NPC in several places, so I was expecting to see the authors applying their NPC to a variety of existing methods in the experimental section, but the paper only experimented with one baseline on each dataset.

[1] Learning with Noisy Correspondence for Cross-modal Matching. NeurIPS2021

[2] Deep Evidential Learning with Noisy Correspondence for Cross-modal Retrieval. ACM MM2022

**Reproducibility:**

4: Could mostly reproduce the results, but there may be some variation because of sample variance or minor variations in their interpretation of the protocol or method.

**Reviewer Confidence:**

5: Positive that my evaluation is correct. I read the paper very carefully and I am very familiar with related work.

**Typos Grammar Style And Presentation Improvements:**

I don't think perplexity can be defined casually since he already has a standard definition in the language modeling field

---

> ### Author Rebuttal · Authors · 2023-08-29
>
> Thanks for your review and suggestions!
>
> > *Q1*: This paper may be the first study of the "mismatched-pair noise" problem in the field of document retrieval and code retrieval. Similar work exists in cross-modal areas.
>
> We appreciate your observation, and indeed, NPC is the first comprehensive study of the "mismatched-pair noise" issue within the domains of document retrieval and code retrieval. Thanks for pointing out these missing cross-modal retrieval works, and we will incorporate discussions with them into the related work section. While the "mismatched-pair noisy" problem in cross-modal retrieval and text retrieval shares similarities, the specific settings and methods used in these two areas are notably distinct.
>
>
> Primarily, the retrieval objective of dense text retrieval is unidirectional, i.e.,  query -> document, while that of cross-modal retrieval is bidirectional, i.e.,  text<->image. Dense text retrieval usually incorporates large document candidate sets for evaluation and hard negative mining, whereas cross-modal retrieval focuses on smaller candidate sets (around 5k for the MSCOCO dataset).
>
> In the area of dense text retrieval, the commonly used method is to employ contrastive loss combined with static global hard negatives mined from large document corpora (about 20 million). On the other hand, cross-modal retrieval commonly employs variations of the triplet loss, which relies solely on in-batch negatives and does not make use of global hard negatives. Moreover, in the context of dense text retrieval, there is a tendency to adopt cross-encoder rankers for distillation, whereas this practice is less common in cross-modal retrieval.
>
> Considering these distinct methodological conventions, it is challenging to directly apply these cross-modal retrieval works to document and code retrieval. We would like to highlight that NPC is the first systematic work to explore mismatched-pair noise in document/code retrieval. NPC is suitable for a variety of existing dense text retrieval methods (i.e., DPR+Hard Negative, coCondenser, RocketQAv2). Our work also encompasses an in-depth analysis, yielding insightful findings. For instance, we recommend the use of random simple negative samples for noise detection, and we highlight the effectiveness of incorporating correction modules to enhance model performance.
>
> We believe that our research and findings on "mismatched-pair noise" are contributing to the development of the community in the direction of text/code retrieval.
>
>
> > *Q2*: NPC does not achieve the purpose of correction. The gap between the NPC results and the DPR-C results is weak, which seems the correction module does not work.
>
> We understand your concern about the effectiveness of the correction module. However, we would like to clarify that the correction module does indeed yield positive results. The specific improvements can be found in Table 3 of our paper's ablation experiments. We can find that the correction module brings consistent improvements, including +2.5 on NQ R@20, +0.89 on NQR@100, +1.0 on StaQC R@1 and +0.96 on StaQC R@3.
>
> We believe that *the weak gap between the NPC results and the DPR-C results* cannot be taken as evidence that *the correction module is ineffective*. The performance difference between NPC and DPR-C can be attributed to two main factors: (1) the negative impact of noise that NPC's noise detection module might have missed, and (2) the positive impact of the correction module. Although NPC's noise detection module is effective, it might not identify every instance of noise, resulting in a small fraction of noise remaining in the training data. Without the correction module, compared to DPR-C trained on clean data, the model's performance could degrade. The enhancement brought by the correction module helps bridge this gap, enabling the model to achieve performance similar to DPR-C. Hence, the correction module is indeed effective.
>
> > *Q3*:  Apply NPC to a variety of existing methods to highlight the plug-and-play nature.
>
> Thank you for raising this point. We sincerely appreciate your suggestion to explore the plug-and-play nature of NPC. Following your suggestion, we conducted comprehensive experiments that integrated NPC into coCondenser and RocketQAv2. The subsequent experiments were conducted on the NQ dataset with a 50% noise ratio.
> We first combined NPC with coCondenser which is a pre-trained model specialized for dense retrieval tasks. The results are shown below.
>
> |      |R@1  | R@5  | R@20 | R@100 |
> | ---- | ---- | ---- | ---- | ---- |
> | coCondenser (BM25 Negative) | 18.28 | 36.37 | 52.01 | 65.92 |
> | coCondenser-C (BM25 Negative) | 44.75     | 68.91 |80.89|87.31     |
> | coCondenser+NPC (BM25 Negative) | 47.31     | 70.38     | 81.58 |87.38 |
>
> |      |R@1  | R@5  | R@20 | R@100 |
> | ---- | ---- | ---- | ---- | ---- |
> | coCondenser (Hard Negative) | 24.55 |44.16 |56.69 |68.72 |
> | coCondenser-C (Hard Negative) | 49.31 | 71.99 |82.41|88.38 |
> | coCondenser+NPC (Hard Negative) | 50.66 |72.42|82.64|88.31  |
>
> From the above tables, it's evident that NPC significantly enhances the robustness of coCondenser against noise associated with mismatched pairs. This observation underscores the compatibility between NPC and pre-trained dense retrievers.
>
> Furthermore, we combined NPC with RocketQAv2 which adopted a cross-encoder as a teacher and dynamically distilled knowledge to the dense retriever. The results are shown below:
>
> |      |R@1  | R@5  | R@20 | R@100 |
> | ---- | ---- | ---- | ---- | ---- |
> | RocketQAv2 | 32.30     | 51.37     | 62.19     |70.79     |
> | RocketQAv2-C | 52.63     | 73.51     | 83.21     |88.71 |
> | RocketQAv2+NPC  | 52.59     | 73.83     | 83.32 |88.69 |
>
> To combine NPC with RocketQAv2, we integrate the noise detection and the correction modules in each training epoch of RocketQAv2. From the table, we can find that although RocketQAv2 uses a powerful cross-encoder as a teacher, it is still limited by the noise of the training data and shows low performance. NPC can effectively harmonize with RocketQAv2 to mitigate the problems caused by mismatched pair noise. We will add these experiment results to the next version.
>
>
>
>
> > *Q4*:  The "warmup" and "iterative detection" are not mentioned before the experiment section.
>
> The "warmup" refers to our initial training stage of the retriever with the traditional contrastive learning method, which provides it with basic retrieval abilities. The "iterative detection" means that we perform noisy detection at each epoch. We will provide additional explanations in Section 3.3 "Overall Procedure" to enhance clarity.
>
> > *Q5*: What is a simple negative sample?  Is a negative sample a simple negative sample if it is not a hard negative sample?
>
> Yes. As explained in line 169 of our paper and further detailed in Appendix C, a "simple" negative document refers to a randomly selected negative document. We will make it clearer in the next version.
>
> > *Q6*: What is the exact expression of $L_{cont}$?
>
> $L_{cont} = - \log p_\theta(d_i^+|q_i,D_{q_{i}}) $
>
> $L_{cont}$ is the contrastive loss, where $d_i^+$ is the annotated positive document of question $q_i$, $D_{q_i}$ is the candidate collection, consisting of an annotated positive document and negative documents.
>
> > *Q7*: Are the results in the paper new SOTA?
>
> The results presented in our paper do not claim to establish new state-of-the-art (SOTA). It's important to note that SOTA methods often leverage techniques such as distillation, data augmentation, and scaling up model sizes to achieve improved performance. Yet, these approaches do not directly align with the core problem we are addressing, making direct and equitable comparisons challenging. In our experiments, we adopt the classical contrastive learning framework as the baseline for a fair comparison.
>
> Moreover, as detailed in our response to Q2, we have compared the compatibility of NPC with more powerful methods (pre-trained retriever coCondenser, RocketQAv2 utilizing cross-encoder distillation), which underscores the effectiveness of our approach. We will add these experiment results to the next version and make the description clearer.
>
> **References**:
>
> [1] RocketQAv2: A Joint Training Method for Dense Passage Retrieval and Passage Re-ranking. EMNLP 2021.
>
> [2] coCondenser: Unsupervised Corpus Aware Language Model Pre-training for Dense Passage Retrieval. ACL 2022.
>
> [3] DPR: Dense Passage Retrieval for Open-Domain Question Answering. EMNLP 2020.

---

### Official Review · Reviewer_7QqM · 2023-08-05

**Soundness:** 3

**Excitement:**

4: Strong: This paper deepens the understanding of some phenomenon or lowers the barriers to an existing research direction.

**Missing References:**

[1] RocketQAv2: A Joint Training Method for Dense Passage Retrieval and Passage Re-ranking. EMNLP 2021.\
[2] LED: Lexicon-Enlightened Dense Retriever for Large-Scale Retrieval. WWW 2023.\
[3] Unifier: A Unified Retriever for Large-Scale Retrieval. KDD 2023

**Paper Topic And Main Contributions:**

This paper proposes a novel framework called Noisy Pair Corrector (NPC), which consists of a noise detection module and a noise correction module. The noise detection module estimates the mismatched pairs in the training data by calculating the perplexity between the annotated positive document and easy negative documents. The noise correction module uses an exponential moving average model to provide rectified soft labels for both noisy and clean data.

**Reasons To Accept:**

- This paper proposes a novel framework called Noisy Pair Corrector (NPC), which consists of a noise detection module and a noise correction module.
- The writing and organization of the paper are well, making it easy to follow and understand.

**Reasons To Reject:**

- The experimental results lack comparison with other state-of-the-art models such as RocketQAv2 [1], LED [2], and Unifier [3].
- The model training used 8 NVIDIA Tesla A100 GPUs and 10 epochs, resulting in high resource consumption.
- The paper lacks experimental results on the MSMARCO, TREC DL 2019, and TREC DL 2020 datasets, which could provide better validation of the model's performance.

[1] RocketQAv2: A Joint Training Method for Dense Passage Retrieval and Passage Re-ranking. EMNLP 2021.\
[2] LED: Lexicon-Enlightened Dense Retriever for Large-Scale Retrieval. WWW 2023.\
[3] Unifier: A Unified Retriever for Large-Scale Retrieval. KDD 2023

**Reproducibility:**

2: Would be hard pressed to reproduce the results. The contribution depends on data that are simply not available outside the author's institution or consortium; not enough details are provided.

**Reviewer Confidence:**

4: Quite sure. I tried to check the important points carefully. It's unlikely, though conceivable, that I missed something that should affect my ratings.

---

> ### Author Rebuttal · Authors · 2023-08-29
>
> Thanks for your valuable comments.
>
> > *Q1*: Lack comparisons with other state-of-the-art models such as RocketQAv2 [1], LED [2], and Unifier [3].
>
>
> To illustrate "mismatched-pair noise" problem more intuitively, we chose the most classical retriever training framework rather than other more complex training frameworks. We would like to highlight that NPC is orthogonal to these state-of-the-art methods. In fact, NPC can be easily applied to these models, helping them to effectively address noise challenges.
>
> Thanks for pointing out the missing related models. The next version will add the discussions with these works. Due to the time limitation, we report the experiment results of RocketQAv2 on the NQ dataset under 20% and 50% noisy ratios. We follow the setting in RocketQAv2, wherein we jointly train a cross-encoder ranker and a dual-encoder retriever. During training, the ranker will dynamically distill knowledge to the dense retriever. To combine NPC with RocketQAv2, we integrate the noise detection and the correction modules in each training epoch of RocketQAv2.
>
> |   NQ Test(20% Noise)   |R@1  | R@5  | R@20 | R@100 |
> | ---- | ---- | ---- | ---- | ---- |
> | RocketQAv2 | 46.62 | 66.24 | 75.63 |82.79 |
> | RocketQAv2+NPC | 55.83| 75.79| 84.68 |89.25 |
>
>
> |   NQ Test(50% Noise)   |R@1  | R@5  | R@20 | R@100 |
> | ---- | ---- | ---- | ---- | ---- |
> | RocketQAv2 | 32.30 | 51.37 | 62.19 |70.79 |
> | RocketQAv2+NPC | 52.59| 73.83| 83.32 |88.69 |
>
> We can see that RocketQAv2+NPC outperforms RocketQAv2. Although RocketQAv2 uses a powerful cross-encoder as a teacher, it is still limited by the noise of the training data and shows low performance. Our NPC can combine this advanced method to mitigate the problems caused by mismatched pair noise.
>
>
>
> > *Q2*: The model training used 8 NVIDIA Tesla A100 GPUs and 10 epochs, resulting in high resource consumption.
>
> The training hyperparameters we adopted are following DPR. We only carry out warmup in the first 10 epochs, and carry out noise detection and correction in the last 30 epochs. Therefore, compared with the classical two-tower model training framework, there is no much higher training cost added. Each epoch takes about 40s on 8\*A100. Note our experiments are adaptable and can be conducted with less demanding hardware resources, should the need arise.
>
> > *Q3*: Experimental results on the MSMARCO, TREC DL 2019, and TREC DL 2020 datasets.
>
> In our initial version, we followed classic DPR to design experiments. As suggested, we have added the experiments on MSMARCO and TREC DL 2019 under a 20% noisy ratio. we implement the classical dense retriever training method, which uses bert-base-uncased mode to initialize our model, using contrast learning combined with one BM25 hard negative to train the dense retriever. The results are shown below:
> |      |MRR@10  | Recall@50  | Recall@1k |  NDCG@10 of TREC-DL-19 |NDCG@10 of TREC-DL-20|
> | ---- | ---- | ---- | ---- | ---- | ---- |
> | DPR (BM25 Negative)  | 23.14     | 65.0     | 85.4     |56.0 | 55.1 |
> | DPR+NPC (BM25 Negative) | 31.93     | 80.6   | 96.4     | 66.5 | 65.6 |
>
> From the table, we can observe that our method NPC still outperforms DPR on the new datasets MSMARCO, TREC DL 2019, and TREC DL 2020. That indicates the effectiveness of NPC in dealing with mismatched pair noise.
>
> > *Q4*: Reproducibility.
>
> We have already uploaded essential code components, and after the anonymous review period, we plan to make the dataset, preprocessing scripts, complete codebase, and model checkpoints publicly available.

---

### Official Review · Reviewer_8vRk · 2023-08-11

**Soundness:** 3

**Excitement:**

3: Ambivalent: It has merits (e.g., it reports state-of-the-art results, the idea is nice), but there are key weaknesses (e.g., it describes incremental work), and it can significantly benefit from another round of revision. However, I won't object to accepting it if my co-reviewers champion it.

**Missing References:**

[1] Learning with Noisy Correspondence for Cross-modal Matching. Proceeding of 35th Conference on Neural Information Processing Systems (NeurIPS 2021).

[2] Learning Cross-Modal Retrieval with Noisy Labels. CVPR 2021.

[3] Noisy Correspondence Learning with Meta Similarity Correction. CVPR 2023.

**Paper Topic And Main Contributions:**

The paper focuses on an important problem in dense retrieval: training an effective model in the presence of mismatched-pair noise. To address this challenge, the authors introduce a novel approach called Noisy Pair Corrector (NPC), which has proven effective in handling both synthetic and realistic noise. The authors validate this approach through experiments on various text-retrieval and code-search benchmarks, such as Natural Question, TriviaQA, StaQC, and SO-DS. The experimental results showcase that NPC achieves good performance, representing an advancement in the field of dense retrieval with noisy data.

**Questions For The Authors:**

A: Is there any direct evidence to support the assumption mentioned in W2, though it is intuitive?

B: What is the difference between the noisy label problem in this study and the ones in [1-3]?

**Reasons To Accept:**

S1: The problem of mitigating the effects of the mismatched-pair noise is very interesting since it's very ubiquitous but hard to deal with.

S2: The proposed method has good mathematical intuition. The design is straightforward and easy to follow.

S3: The experimental results are pretty good compared to the generally comprehensive baselines.

**Reasons To Reject:**

W1: Similar ideas have already been adopted in cross-modal retrieval tasks, like [1-3] listed in the missing reference, which hurts the novelty of the paper.

W2: The assumption in section 3.1 : "dense retrievers will first learn to distinguish correctly matched pairs from easy negatives, and then gradually overfit the mismatched pairs" needs to be explained more thoroughly by some specific empirical analysis, since it serves as the basis of this study.



**Reproducibility:**

4: Could mostly reproduce the results, but there may be some variation because of sample variance or minor variations in their interpretation of the protocol or method.

**Reviewer Confidence:**

3: Pretty sure, but there's a chance I missed something. Although I have a good feel for this area in general, I did not carefully check the paper's details, e.g., the math, experimental design, or novelty.

---

> ### Author Rebuttal · Authors · 2023-08-29
>
> Thanks for your valuable comments.
>
> > *Q1*:  Is there any direct evidence to support the assumption "dense retrievers will first learn to distinguish correctly matched pairs from easy negatives, and then gradually overfit the mismatched pairs".
>
> We are grateful for your insightful question. In Figures 4 of our paper, we present evidence that addresses your concern.
>
> Figure 4.a and Figure 4.b illustrate the perplexity distributions for both correctly matched and mismatched query-document pairs before and after the warmup stage, respectively. We can find that Figure 4.a shows a narrow divergence between the perplexity distribution of correctly matched pairs and that of mismatched pairs, indicating the model can not distinguish them before training. After a few epochs, a remarkable shift becomes apparent. In Figure 4.b, the perplexity of correctly matched pairs experiences a rapid decrease, signifying the model's improving ability to distinguish them accurately. In contrast, the perplexity of mismatched pairs exhibits a contrasting trend. These empirical findings support our assumption *"dense retrievers will first learn to distinguish correctly matched pairs from easy negatives"*.
>
> Furthermore, Figure 4.c shows the perplexity distribution of query-document pairs when we continue training the retriever with the original contrastive learning objective. We can find that the perplexity of mismatched pairs is notably reduced compared to that in Figure 4.b. More remarkably, the perplexity of mismatched pairs is reduced to a range that is indistinguishable from correctly matched pairs. This phenomenon intuitively supports our assumption *"then gradually overfit the mismatched pairs"*.
>
>
> >*Q2*:  Similar ideas in cross-modal retrieval area[1-3].
>
> Thanks for pointing out these missing cross-modal retrieval works, and we will incorporate discussions with them into "Related Work" section. Although the "mismatched-pair noisy" problem in cross-modal retrieval and text retrieval shares similarities, the specific settings and methods used in these two areas are notably distinct.
>
> Primarily, the retrieval objective of dense text retrieval is unidirectional, i.e.,  query -> document, while that of cross-modal retrieval is bidirectional, i.e.,  text<->image. Dense text retrieval usually incorporates large document candidate sets for evaluation and hard negative mining, whereas cross-modal retrieval focuses on smaller candidate sets[1][2][3] (around 5k for the MSCOCO dataset).
>
> In the area of dense text retrieval, the commonly used method is to employ contrastive loss combined with static global hard negatives mined from large document corpora (about 20 million). On the other hand, cross-modal retrieval commonly employs variations of the triplet loss[1][2][3], which relies solely on in-batch negatives and does not make use of global hard negatives. Moreover, in the context of dense text retrieval, there is a tendency to adopt cross-encoder rankers for distillation, whereas this practice is less common in cross-modal retrieval.
>
> Considering these distinct methodological conventions, it is challenging to directly apply these cross-modal retrieval works to document and code retrieval. We would like to highlight that NPC is the first systematic work to explore mismatched-pair noise in document/code retrieval. NPC is suitable for a variety of existing dense text retrieval methods (i.e., DPR+Hard Negative, coCondenser, RocketQAv2), specific results can be found in our reply Reviewer#3 Q3. Our work also encompasses an in-depth analysis, yielding insightful findings. For instance, we recommend the use of random simple negative samples rather than hard negatives for noise detection. We highlight the effectiveness of incorporating correction modules to enhance model performance.
>
> We believe that our research and findings on “mismatched-pair noise” are contributing to the development of the community in the direction of text/code retrieval.

---

### Meta-Review · Area_Chair_FW47 · 2023-09-20

**Recommendation:** 3

**Metareview:**

This submission studies the noisy label issue in dense retrieval, which is neglected in the existing work. However, the technical novelty of the proposed method is limited since similar technique has been proposed in similar IR tasks. While the rebuttal has highlighted some differences between the proposed method and existing methods, these points are minor, making this work a more incremental one.

---

### Decision · Program_Chairs · 2023-10-07

**Decision:**

Accept-Findings

**Comment:**

This submission studies the noisy label issue in dense retrieval, which is neglected in the existing work. However, the technical novelty of the proposed method is limited since similar technique has been proposed in similar IR tasks. While the rebuttal has highlighted some differences between the proposed method and existing methods, these points are minor, making this work a more incremental one.